# Retinal Neurodegeneration in an Intraocular Pressure Fluctuation Rat Model

**DOI:** 10.3390/ijms25073689

**Published:** 2024-03-26

**Authors:** Jeong-Sun Han, Chan Kee Park, Kyoung In Jung

**Affiliations:** Department of Ophthalmology, Seoul St. Mary’s Hospital, College of Medicine, The Catholic University of Korea, Seoul 06591, Republic of Korea; winehan@catholic.ac.kr (J.-S.H.); ckpark@catholic.ac.kr (C.K.P.)

**Keywords:** gliosis, intraocular pressure fluctuation, neurodegeneration, oxidative stress, retinal ganglion cells

## Abstract

Increased intraocular pressure (IOP) is the most important risk factor for glaucoma. The role of IOP fluctuation, independently from elevated IOP, has not yet been confirmed in glaucoma. We investigated the effects of IOP fluctuation itself on retinal neurodegeneration. Male rats were treated with IOP-lowering eyedrops (brinzolamide and latanoprost) on Mondays and Thursdays (in the irregular instillation group) or daily (in the regular instillation group), and saline was administered daily in the normal control group for 8 weeks. The IOP standard deviation was higher in the irregular instillation group than the regular instillation group or the control group. The degree of oxidative stress, which was analyzed by labeling superoxide, oxidative DNA damage, and nitrotyrosine, was increased in the irregular instillation group. Macroglial activation, expressed by glial fibrillary acidic protein in the optic nerve head and retina, was observed with the irregular instillation of IOP-lowering eyedrops. Microglial activation, as indicated by Iba-1, and the expression of TNF-α did not show a significant difference between the irregular instillation and control groups. Expression of cleaved caspase-3 was upregulated and the number of retinal ganglion cells (RGCs) was decreased in the irregular instillation group. Our findings indicate that IOP fluctuations could be induced by irregular instillation of IOP-lowering eyedrops and this could lead to the degeneration of RGCs, probably through increased oxidative stress and macrogliosis.

## 1. Introduction

Glaucoma is a major disease that can lead to irreversible vision loss [1]. Elevated IOP is a major causative factor in the development and progression of glaucoma [2,3]. However, glaucoma develops within the statistically normal range of IOP in normal-tension glaucoma (NTG) and could progress despite relatively low IOP values with treatment [3]. IOP is not a constant value and shows variations during the day/night cycle or longer periods [4]. Patients with glaucoma, including NTG, have shown greater IOP fluctuations over the course of a day than healthy control subjects [5,6,7,8,9]. Compared to elevated IOP values, the role of IOP fluctuation itself has been less clearly established in glaucomatous damage because existing studies have argued for and against the effects of IOP fluctuation in glaucoma development or progression [10,11,12,13]. In the Advanced Glaucoma Intervention Study (AGIS), long-term IOP fluctuation was related to visual field progression only in glaucoma patients with low mean IOP values [14]. Recently, IOP fluctuation was found to be a significant prognostic factor associated with the progression of glaucoma, especially in patients with NTG, even when the IOP is in the low teens [15,16,17]. In patients with high IOP values, it is understandable that IOP fluctuation could be related to intermittent high peak IOP values, which could result in the progression of glaucoma. In glaucoma patients with IOP values in the low teens, however, the significant effects of IOP fluctuation on the fast progression of glaucoma are not easily predictable.

A positive correlation between IOP fluctuation and hemodynamic blood pressure variability has been reported in patients with NTG and IOP values in the low teens [17]. Our group and others have reported that patients with NTG have a higher probability of having abnormal autonomic dysfunction as analyzed by heart rate variability [18,19,20]. Blood flow instability is also a risk factor for glaucoma development and progression [21,22]. Greater IOP fluctuation could be hypothesized to affect the progression of glaucoma more strongly when combined with blood flow dysregulation. However, no study has evaluated the relationship between IOP fluctuation itself and the loss of retinal ganglion cells (RGCs) apart from systemic blood flow instability. In clinical human studies, it is difficult to exclude the systemic effects of blood pressure or the autonomic nervous system completely when analyzing the effects of IOP fluctuation on glaucoma. It is necessary to conduct a prospective functional study where IOP fluctuation is aggravated or attenuated in order to reveal the effects of IOP fluctuation itself on glaucomatous optic nerve damage. 

There is no known animal model able to support an investigation of the relationship between IOP fluctuation and glaucomatous damage within the normal range of IOP. One animal study induced daily 1 h IOP elevations using a vascular loop and demonstrated RGC loss [23]. In that study, IOP fluctuation was established by intermittently increasing the IOP to 35 mmHg [23]. It was not clear whether the axonal loss of RGCs was caused by short-term IOP elevation or the IOP fluctuation itself.

Given these findings, animal models free from systemic diseases, in which the IOP is not increased above normal levels are required to assess the effects of IOP fluctuation on RGC damage. In this study, we established an IOP fluctuation model induced by the instillation of IOP-lowering eyedrops intermittently in ordinary rats in which the IOP was within the normal range. The effects of the IOP fluctuation itself on RGC loss and the mechanisms at play were investigated.

## 2. Results

### 2.1. IOP Fluctuation Induced by Irregular Instillation of IOP-Lowering Eyedrops

The average IOP was lower in the irregular instillation group (9.6 ± 0.4 mmHg) and the regular instillation group (9.5 ± 0.4 mmHg) than in the control group (10.1 ± 0.4 mmHg, *p* < 0.001, Figure 1A). The daily IOP measurement standard deviation was greater in the irregular instillation group (0.8 ± 0.1 mmHg) than in the regular instillation group (0.5 ± 0.1 mmHg) or the control group (0.6 ± 0.3 mmHg, *p* < 0.001, Figure 1B).

### 2.2. Oxidative Stress and DNA Damage

Red fluorescence on dihydroethidium (DHE) staining, which detects superoxide, was scant in the normal control group (Figure 2A,B). In the irregular instillation group, DHE expression was increased from the ganglion cell layer (GCL) to the outer nuclear layer overall compared to the control group or the regular instillation group (*p* < 0.001). Immunofluorescence staining for nitrotyrosine, an indicator of nitro-oxidative stress, was nearly absent in the normal control group (Figure 2C,D). In the irregular instillation group, nitrotyrosine expression was upregulated in the GCL, inner plexiform layer, and outer plexiform layer compared to both the normal control group and the regular instillation group (*p* < 0.001). 8-OHdG expression, a marker of oxidative DNA damage, was rare in the control retina (Figure 3A,B). In the irregular instillation group, 8-OHdG immunostaining was elevated, especially in the GCL, compared to in the normal control group or the regular instillation group (*p* < 0.001). Immunostaining for γH2AX, which is an indicator of DNA damage (especially double-strand breaks) [24], was observed rarely in the GCL in the normal control group (Figure 3C,D). In the irregular instillation group, the number of cells labeled with the γH2AX antibody was increased in the GCL compared to in the normal control group or the regular instillation group (*p* = 0.014).

### 2.3. Mitochondrial Protein for Oxidative Phosphorylation

Among oxidative phosphorylation complexes I–V in mitochondrial samples isolated from the retina, bands for complexes I, III, IV, and V were detected (Figure 4A). The expression levels of complexes III and V (ATP synthase) were lower with the irregular instillation of IOP-lowering eyedrops than with the regular instillation of them (Figure 4C,E) (*p* = 0.043 and *p* = 0.025, respectively). However, a significant difference between the irregular instillation group and the normal control group was not found by post hoc tests.

### 2.4. Macrogliosis

In vertical retinal sections, glial fibrillary acidic protein (GFAP) immunostaining was limited to astrocytes in the retinal nerve fiber layer and the end-feet of Müller cells at the internal limiting membrane in the normal control group (Figure 5A,B). In the irregular instillation group, GFAP expression by immunofluorescence staining was extended to the inner plexiform layer and greater than that in the control group (*p* = 0.045). Kir 4.1 expression on immunofluorescence staining was observed broadly from the end-feet of Müller cells from the internal limiting membrane to the outer plexiform layer and around the vessels in the normal control group (Figure 5C,D). In the irregular instillation group, Kir 4.1 labeling was reduced significantly overall compared to that in the control group or the regular instillation group (*p* < 0.001). In the optic nerve head (ONH), GAFP immunofluorescence staining was relatively weak in the normal control group and increased with the irregular instillation of IOP-lowering eyedrops (*p* = 0.001; Figure 5E,F). In the irregular instillation group, GFAP expression was predominant in the glial lamina region, suggesting the activation of astrocytes in the ONH. GFAP labeling in the ONH was negatively correlated with the proportion of RGCs stained with Brn3a in the retina (r = −0.245, *p* = 0.027, Appendix A).

### 2.5. Microgliosis

Expression of Iba-1, a surrogate marker of activated microglia, showed weak and small cell bodies mainly in the inner plexiform layer in the control group by immunofluorescence staining (Figure 6A). Immunostaining of Iba-1 increased, showing enlarged soma of microglia, with the irregular instillation of IOP-lowering eyedrops, but there was no significant difference between groups (Figure 6B). Western blotting confirmed that Iba-1 expression was greater in the irregular administration group than in the regular administration group and did not significantly differ from that in the control group (*p* = 0.014, the irregular instillation group > the regular instillation group by post hoc analysis, Figure 6C). In terms of Iba-1 immunostaining, the ONH exhibited upregulated expression in the irregular instillation group, but there was no statistically significant difference between groups (*p* = 0.546, Figure 6D,E). Western blotting of the optic nerve showed increased expression of Iba-1 in the irregular instillation group compared to the other groups, but statistical analysis could not be completed because the optic nerve samples were few in number and were combined for each group (Figure 6F). P2Y12 expression by immunofluorescence staining showed scant fluorescence in the GCL and inner nuclear layer in the normal control group (Appendix A). In the irregular instillation group, P2Y12 immunofluorescence staining was increased, especially in the GCL and inner nuclear layer, compared to in the normal control group or the regular instillation group, but there was no significant difference between groups (Appendix A). Western blot analysis demonstrated the presence of upregulated P2Y12 expression in the irregular instillation group compared to the regular instillation group, but there was no statistically significant difference from the normal control group (*p* = 0.017; irregular instillation group > regular instillation group by post hoc analysis, Appendix A). Immunoblot staining for TNF-α did not reveal a significant difference between the groups (*p* = 0.303, Figure 6G,H).

### 2.6. Degeneration of RGCs

Immunofluorescence staining of cleaved caspase-3, a key mediator of apoptosis, did not show positive staining in the normal control group (Figure 7A,B). In the irregular instillation group, the labeling of cleaved caspase-3 was observed in a few cells in the GCL. The number of cleaved caspase-3-positive cells was greater in the irregular instillation group than in the normal control group or the regular instillation group (*p* = 0.014). Expression of Brn-3a, which labels RGCs, was found in the GCL in the normal control group (Figure 7C,D). In the irregular instillation group, the number of Brn3a-positive cells was lower than that in the normal control group or the regular instillation group (*p* = 0.002). Western blotting of Bax, an apoptotic marker, showed elevated expression in the irregular instillation group compared to the regular instillation group (*p* = 0.014, Figure 7E).

### 2.7. Ultrastructural Findings of the Optic Nerve

The normal control group showed a normal axoplasm enclosed by well-organized myelin sheathing. In the irregular instillation group, the myelin sheath was delaminated and the axoplasm was sparse compared to in the control group or the regular instillation group (Appendix A).

## 3. Discussion

This study demonstrated that increased IOP fluctuation within the normal range of IOP could be induced by intermittent application of IOP-lowering eyedrops in rats. Eyes with greater IOP fluctuation showed more oxidative stress and increased macrogliosis in the retina and ONH. Eyes with greater IOP fluctuation were also found to have decreased numbers of RGCs with concomitant elevated cleaved caspase-3 levels when compared to normal control eyes. 

Both intermittent and regular daily instillation of IOP-lowering eyedrops decreased mean IOP compared to the IOP values of the control group, and there was no significant difference in mean IOP values between the irregular and regular instillation groups. Irregular instillation of IOP-lowering eyedrops increased IOP standard deviation, inducing larger IOP fluctuations than those seen in the control group or regular instillation group. Regarding the effects of IOP reduction by topical glaucoma medication in normal rats, the regular instillation group did not present a difference in the number of RGCs compared to the control group. Therefore, we could assume that greater IOP fluctuation, rather than IOP reduction or medication administration, relates to RGC loss.

Our study demonstrated that an IOP-fluctuation model could be induced by intermittent instillation of IOP-lowering eyedrops in the normal range of IOP. Previously, a study detected the axonal loss of RGCs by inducing daily 1-h IOP elevations over 6 weeks using a vascular loop strategy in rats [23]. In that study, intermittent IOP elevation to 35 mmHg induced a reduction in both the retinal nerve fiber layer thickness and the number of cell soma in the GCL [23]. However, that study did not prove that RGC degeneration was caused by larger IOP fluctuations because RGC loss can be triggered not only by IOP fluctuation but also by IOP elevation even when the increased IOP is maintained for a short time. Therefore, this study is the first to report that IOP fluctuation itself, not IOP elevation, could be associated with RGC loss using a rodent model.

Regarding oxidative stress, expression levels of superoxide, as a reactive oxygen species; nitrotyrosine, which is thought to be a detectable biomarker of peroxynitrite formation; and 8-OHdG, which is an indicator of oxidative DNA damage, were increased in the irregular instillation group compared to the control group and the regular instillation group. Accumulating evidence highlights that oxidative stress is associated with and contributes to neurodegeneration in glaucoma [25,26]. Liu et al. previously reported that oxidative stress is an early event in hydrostatic pressure or IOP-induced RGC loss [25]. We found that greater IOP fluctuations, even in the normal range, are associated with greater oxidative stress. The mechanical stress from IOP fluctuation could affect the RGCs and glial cells and capillaries, especially in the glial lamina, which is a weak point of the outer layer of the eyeball [27]. Consequently, increased reactive oxygen species from cells placed under mechanical stress and impaired blood flow or ischemic–reperfusion injury might induce oxidative stress, even though the exact mechanism of how greater IOP fluctuations trigger oxidative stress has not yet been confirmed [4,27].

Mitochondria are the main source of reactive oxygen species in the body because the electron transport chain in mitochondria uses about 85% of the oxygen that a typical cell consumes [28,29]. Regarding the isolated retinal mitochondrial samples, greater IOP fluctuations showed a tendency to decrease the expression of oxidative phosphorylation complexes III and V, but there was no significant difference between the irregular instillation group and the control group. We assumed that a smaller number of mitochondrial samples (eight samples per group) resulted in the non-significant difference because mitochondrial isolation from retinal tissue was performed independently from samples used for Western blotting or immunofluorescence staining using other antibodies. Besides the mitochondria electron transport chain, other sources, such as the endoplasmic reticulum and peroxisomes, may generate the reactive oxygen species induced by IOP fluctuation-related stress [28].

Our study also revealed macroglial activation in the retina and ONH after the induction of IOP fluctuation. Larger IOP fluctuations induced by IOP-lowering eyedrops could place intermittent mechanical stress on the retina or ONH. Increased expression of intermediate filaments like GFAP, which is known to be the most non-specific sensitive indicator of retinal stress or injury [30], represents reactive gliosis in the retina and ONH. Previously, expression of GFAP was found relatively early at 2 h after the induction of chronic IOP elevation in rats [31]. It is not clear whether gliosis is neuroprotective or detrimental to neurons, but long-term gliosis seems to be related to glial scar formation and could be harmful to neurons [32]. In our study, immunofluorescence staining of anti-Kir 4.1 antibodies was localized in accordance with Müller cells in the normal retina. Kir 4.1 expression was attenuated in the larger IOP fluctuation group compared to the control group. Kir 4.1 is one of the inwardly rectifying Kir channels responsible for maintaining membrane potential and extracellular potassium concentrations [33]. Downregulation of Kir 4.1 in the retina has been reported in chronic ocular hypertension models, although there were no significant changes in membrane potassium channels in DMB/2J mice [34,35,36]. Müller cell gliosis induced by IOP fluctuation could indicate a reduced ability of Müller cells to maintain electrolyte homeostasis or membrane potential in the retina [34].

In the ONH, GFAP immunostaining increased in the irregular instillation group, which demonstrated reactive and hypertrophied astrocytes [37,38,39]. In rodent models with chronic ocular hypertension or human glaucoma eyes, GFAP expression was found to increase in the ONH, but the results vary depending on the duration of IOP elevation or location of the ONH [40,41,42]. Joos et al. reported that intermittent IOP elevation of >30 mmHg induced increased GFAP expression in the optic nerve and axonal loss, corresponding to the results of our study [23]. In the irregular instillation group, the study observed both macroglial activation and a reduction in the number of RGCs. This suggests a potential association between gliosis and RGC loss. To quantitatively assess this relationship, we measured the correlation between the degree of GFAP fluorescence staining and the number of Brn3a-positive RGCs. The degree of immunofluorescence staining of GFAP in the ONH was negatively correlated with the number of RGCs. The significant negative correlation between the macrogliosis in ONH and the number of RGCs indicates that alteration of the ONH microenvironment by reactive gliosis induced by IOP fluctuation might contribute to the axonal degeneration associated with disturbed axonal transport, leading to degeneration of RGC soma—although it has not been established whether astrogliosis is the aggravating factor or the result of neurodegeneration [40,42]. Further functional studies are warranted to confirm the role of astrogliosis in the neurodegeneration induced by IOP fluctuation.

Regarding microgliosis, immunofluorescence staining and Western blotting for anti-Iba-1 antibody, which has been used as a microglial marker, revealed an increased expression pattern with enlarged soma in the retina and ONH in the larger IOP fluctuation group, but there was no significant difference in findings compared to those of the control group. These tendencies align with previous studies which have reported larger cell bodies, an increased area of Iba1-positive cells, and/or a retracted and thickened microglial processes in the chronic ocular hypertension group or optic nerve crush models [43,44,45]. Our study used only vertical sections of the retina for microglia evaluation, while several other studies employed flat mounts of the retina to assess microglial processes stained with Iba-1 antibody [43,44,45]. This methodological difference may have influenced our observations, as the delicate staining of microglial branches might not be fully captured in vertical retinal sections, especially in the normal control group where thin processes of microglia might not be visible. The primary step of microglial activation is upregulation of P2Y12, which is greatly labeled in activated non-inflammatory M2 microglia that have been known to be activated in the initial stages after IOP increase [46,47]. In our study, there was no significant difference in P2Y12 expression, as revealed by Western blotting, between the IOP fluctuation group and the control group. It has been reported before that microglial activation is followed by the production of pro-inflammatory factors such as TNF-α [37]. We found that TNF-α expression in the retina was not increased by the induction of IOP fluctuation. IOP fluctuation within normal IOP levels did not seem to induce significant microglial activation or inflammation, although chronic IOP elevation above normal levels could result in microglial activation and neuroinflammation [37].

Expression of cleaved caspase 3, an apoptosis marker, was increased and the number of RGCs stained by Brn3a was decreased. We speculate that the primary mechanism underlying RGC loss in the irregular instillation group was from increased IOP fluctuation. Greater IOP fluctuation could activate mechanosensitive macroglial cells such as astrocytes or Müller cells [48,49]. Our study observed macroglial activation and signs of glial dysfunction, including disturbed potassium homeostasis. Müller cells, known for their role in the defending against oxidative stress through production of the antioxidant glutathione, may play a pivotal role in this process [50]. Dysfunction of Müller cells could lead to oxidative stress, contributing to the neurodegeneration of RGCs. Thus, gliosis associated with glial dysfunction and elevated oxidative stress could potentially influence RGC loss in the irregular instillation group, even though it has not been made clear yet how gliosis or oxidative stress is associated with RGC loss [28]. Further studies are required to investigate the specific mechanism behind how and in what order IOP-related stress (e.g., greater IOP fluctuation) affects RGCs, glial cells, and blood vessels.

In our study, we observed statistically significant elevations in oxidative stress-markers such as dihydroethidium (DHE) and 8-hydroxy-2′-deoxyguanosine (8-OHdG), in the regular instillation group compared to the control group. This finding aligns with previous research by Sedlak et al., who reported increased oxidative stress in the tear film following topical administration of prostaglandin analogues, including latanoprost [51]. The role of prostaglandin F2 alpha as biomarkers of oxidative stress in systemic diseases has been also suggested [52]. Topical application of latanoprost can cause cystoid macular edema in the retina, indicating a potential effect on the posterior segment. Despite the observed increase in oxidative stress associated with latanoprost, our study did not find evidence of RGC loss in our study. Additionally, while oxidative stress was slightly elevated in the regular instillation group, it was markedly greater in the irregular instillation group. This suggests that the greater IOP fluctuation induced by irregular instillations may have a more pronounced effect on oxidative stress and RGC loss compared to the eyedrop itself. Topical instillation of latanoprost has been the most popular first line treatment and has demonstrated superior efficacy in lowering IOP in patients with NTG [53,54]. We hypothesized that NTG patients with weaker supportive tissue such as thinner lamina cribrosa thickness or vascular instability, may derive greater benefit from stable IOP reduction through instillation of antiglaucoma eyedrops, even considering the slight elevation in oxidative stress.

In diabetic rats, we previously reported that early loss of RGCs may be associated with higher IOP fluctuations and slightly elevated IOP values within the normal range, an effect that was relieved by IOP-lowering eyedrops [55]. In terms of IOP, both slightly increased IOP values and IOP fluctuations might affect RGC loss in the diabetic retina. Diabetes is associated with autonomic dysfunction and a low potential to autoregulate blood flow [56]. In the previous study, we also could not exclude the possibility that RGC degeneration in the diabetic retina could result not only from IOP fluctuation but also IOP fluctuation combined with slightly increased IOP or blood flow instability [55]. In the current study, however, we demonstrated that IOP fluctuation itself could be related to RGC degeneration independent of systemic blood flow disturbance because normal rats without disease were used. In addition, we selected latanoprost and brinzolamide eyedrops as IOP-lowering medications because other glaucoma medications like β-blocker or α2 agonists could reduce blood pressure [57].

Access to appropriate animal models for glaucoma is essential to elucidate the pathogenesis of this disease and establish superior treatments. So far, the most existing animal models for glaucoma have focused on chronic IOP elevation by episcleral cauterization, intracameral injection of microbeads, or circumlimbal suture [58]. Several methods to establish pressure-independent animal models of glaucoma have been suggested, such as optic nerve transection, optic nerve crush, or intravitreal injection of excitotoxic amino acids [58]. The optic nerve crush model resembles traumatic optic neuropathy, which is clinically different from glaucoma because optic disc pallor is more prominent than optic disc cupping. The role of excitotoxicity in glaucoma has not yet been confirmed, although excitotoxicity might be involved in the pathogenesis of glaucoma [58]. In addition, those animal models did not present the feature of glaucomatous axonal loss, such as alterations of the lamina cribrosa or glial lamina, even though a loss of RGCs was shown.

Regarding IOP fluctuation, clinical evidence supports the concept that IOP fluctuation might play an important role in glaucoma, including NTG [5,6,7,8,9]. In our study, RGC loss was found in the IOP fluctuation rat model. ONH analysis revealed gliosis in the glial lamina and the correlation between macrogliosis in the ONH and RGC loss. Therefore, the IOP-fluctuation model proposed in this study could be used as an NTG model. IOP fluctuation was induced by the treatment of IOP-lowering eyedrops and the mean IOP was lower in the IOP fluctuation group than in the control group. The effects of greater IOP fluctuation combined with a low mean IOP value might differ from the effects of IOP fluctuation in patients with high-tension glaucoma, but might be similar to those found in patients with NTG. We are planning a future study to investigate the effects of IOP fluctuation in a chronic ocular hypertension model to evaluate the effects of IOP fluctuation at high IOP levels on RGC loss.

It is intuitive that poor medication adherence leads to poor outcomes in the management of glaucoma. Several clinical studies found that poor adherence is associated with faster glaucoma progression [59,60]. The mechanism by which low adherence could affect more rapid glaucoma progression might involve inadequate IOP control and/or IOP fluctuation [60]. Our study found that IOP fluctuation induced by irregular medication application, skipping doses more than two thirds of the time, resulted RGC loss. Our study regimen corresponds to very poor medication adherence, and our results align with clinical studies reporting that patients who missed medication doses at two-thirds of visits showed faster glaucoma progression than those reporting missed doses at one-third to two-thirds of visits [59]. Our study suggests that very poor adherence might induce greater IOP fluctuation, offsetting the mean reduction of IOP achieved by medication. Through this study, showing that irregular treatment with glaucoma eyedrops induced RGC loss in normal rats, the emphasis on regular instillation of IOP-lowering eyedrops cannot be overstated to patients with glaucoma, even those with NTG.

## 4. Materials and Methods

### 4.1. Experimental Animals

Adult male Sprague–Dawley rats (7–8 weeks old, 200–300 g) were employed according to the Association for Research in Vision and Ophthalmology Statement on the Use of Animals in Ophthalmic and Vision Research. All animal research procedures were allowed by the Institutional Animal Care and Use Committee of the School of Medicine of The Catholic University of Korea and the Institutional Animal Care and Use Committee of the Department of Laboratory Animals at The Catholic University of Korea, Songeui Campus.

### 4.2. Allocation of Groups and Drug Treatments

The rats were allocated to the irregular instillation group, regular instillation group, or control group. IOP-lowering eyedrops were administered to rats on Mondays and Thursdays in the irregular instillation group and daily in the regular instillation group. (Appendix A). Meanwhile, saline was given to the control group daily. Among the groups, rats were given either a single drop of 1% brinzolamide ophthalmic suspension (Azopt^®^; Alcon Laboratories, Inc., Ft. Worth, TX, USA) twice a day at about 10:00 a.m. and 6:00 p.m., 0.005% latanoprost ophthalmic solution (Xalatan^®^; Pfizer, New York, NY, USA) once a day between 10:00 a.m. and 12:00 p.m., or saline once a day between 10:00 a.m. and 12:00 p.m. for 8 weeks in both eyes. Eighteen rats were randomly allocated to each group. For the analysis of mitochondrial proteins, eight rats were assigned to each group.

### 4.3. Measurement of IOP

IOP was assessed under general anesthesia using isoflurane–nitrous oxide at baseline and every day between 10:00 a.m. and 12:00 p.m. before applying the drugs to each animal by using a rebound tonometer (TonoVet^®^; Icare, Helsinki, Finland). IOP was measured five times. Averages of five IOP measurements were calculated every day for each eye. The SD of the mean IOP values over the course of 8 weeks for each eye was then calculated. For each experimental group, the mean and SD of each SD measurement in rats included in each experimental group were determined.

### 4.4. Immunofluorescence Staining

The enucleation of eyeballs was completed after sacrificing rats by CO2 inhalation. Enucleated eyes were washed with phosphate-buffered saline (PBS). Eyeballs were immersed in 4% paraformaldehyde in 0.1 M of phosphate buffer for 20 min. After removing the anterior segments of the eyes, the posterior eyecups were immersed in 4% para-formaldehyde in 0.1 M of phosphate buffer for 1 h at 4 °C, then rinsed with PBS and immersed in 0.1 M of phosphate buffer including 25% sucrose at 4 °C overnight. After rinsing with PBS, the samples were exposed to an optimal cutting temperature compound and frozen with liquid nitrogen. Cryostat sections (12 µm) were made and kept at −20 °C.

Cryosections were thawed, air-dried, and rinsed with PBS, then treated with 3% Triton X-100 for 30 min and exposed to 10% normal donkey serum for 1 h. Next, the slides were treated with anti–GFAP (1:400, #MAB360; Millipore, Burlington, MA, USA), anti-Kir 4.1 (1:500, #APC-035; Alomone Labs, Jerusalem, Israel), anti-ionized calcium binding adaptor molecule (Iba1, 1:500; Wako Pure Chemical Industries Ltd., Osaka, Japan), anti-P2Y12 (1:100, #APR-020; Alomone), nitrotyrosine (1:10, #MBSB0172; MyBioSource, Inc, San Diego, CA, USA), 8-hydroxy-2′-deoxyguanosine (8-OHdG) (1:500, #ab62623; Abcam, Cambridge, UK), anti-γH2AX (phosphoS139, #ab81299, 1:200; Abcam, Cambridge, UK), anti-cleaved caspase-3 (1:100, #9661S; Cell Signaling Technology, Danvers, MA, USA), and anti-Brn3a (1:200, #SC-8429; Santa Cruz Biotechnology, Dallas, TX, USA) overnight at 4 °C. The sections were then incubated with Alexa Fluor 488-labeled goat anti-mouse immunoglobulin G (#A-11001; Thermo Fisher Scientific, Waltham, MA, USA) or Alexa Fluor 546-labeled goat anti-rabbit immunoglobulin G (A-11010; Thermo Fisher Scientific, Waltham, MA, USA) for 1 h. Next, the sections were mounted using VECTASHIELD mounting medium with DAPI (Vector Laboratories, Burlingame, CA, USA). All images were analyzed using ImageJ (version 1.40; U.S. National Institute of Health, Bethesda, MD, USA). The collected images were analyzed after each eyecup was divided into 2 mid-central areas (approximately 1.5 mm from the optic nerve) and 2 peripheral areas (approximately 3.5 mm from the optic nerve) (Appendix A). The multicolor images were divided into separate channels, which were converted to grayscale before processing. We assessed fluorescence levels higher than a threshold using the “set measurements” tool. To accurately quantify the expression levels of DHE, Nitrotyrosine, 8-OHdG, GFAP, Kir4.1, and Iba1, the polygon selection tool in ImageJ software was utilized. This tool facilitated the delineation of specific retinal regions, spanning from the GCL to the outer nuclear layer (ONL). For a thorough and representative analysis of the retinal architecture, measurements were conducted across both midline and peripheral sections, with two of each being analyzed. The mean fluorescence intensity within this delineated area was computed to quantify the expression of the specified markers.

### 4.5. Measurement of Reactive Oxygen Species

The retinal sections were incubated with 5 µM of DHE (D11347; Invitrogen, Carlsbad, CA, USA) for 20 min per the manufacturer’s recommendations. DHE is converted to an ethidium derivative when it reacts with intracellular superoxide [61], which then binds to the deoxyribonucleic acid, thereby enabling cells to emit red fluorescence [62]. The samples were photographed at an excitation wavelength of 520 nm and an emission wavelength of 610 nm.

### 4.6. Mitochondrial Isolation

The mitochondria proteins were isolated with a mitochondria-isolation kit for retina tissue (#89801; Thermo Fisher Scientific, Waltham, MA, USA) according to the manufacturer’s instructions. A handy vortex mixer (#Z359971; Sigma-Aldrich, St. Louis, MO, USA) was utilized to mechanically disrupt cells for mitochondria isolation. The tissue homogenates were then subjected to 3000× *g* for 15 min at 4 °C. The supernatants were then centrifuged at 12,000× *g* for 5 min to obtain the mitochondrial pellet. The concentration of mitochondrial protein was measured using a bicinchoninic acid protein assay kit (#PI23235; Thermo Fisher Scientific, Waltham, MA, USA).

### 4.7. Western Blot Analysis

Protein extraction and Western blotting were completed as previously stated [31]. The retinal tissues were lysed in radioimmunoprecipitation assay buffer, and the total protein value was analyzed using a standard bicinchoninic acid assay (Thermo Fisher Scientific, Darmstadt, Germany). The sample buffer was administered to the retinal tissue, including 30 μg of total protein. The protein was separated by 10% sodium dodecyl sulfate–polyacrylamide gel electrophoresis and affixed to a nitrocellulose membrane. The membranes were rinsed and treated with 5% skim milk in Tris-buffered saline/Tween 20 (TBST) buffer for 1 h at room temperature. Then, the membranes were treated with the following antibodies: anti-Iba-1 (1:1000; Ako Pure Chemical), anti-P2Y12 (1:1000, #APR-020; Alomone), anti-Bax (1:1000, #ab32503; Abcam, Cambridge, UK), anti–TNF-α (1:1000, #ab1793; Abcam, Cambridge, UK), anti-phospho-p38MAPK 1:1000, (#9211; Cell Signaling Technology, Danvers, MA, USA), anti-JNK (1:1000, #9251; Cell Signaling Technology, Danvers, MA, USA), and actin (1:200, #sc-47778; Santa Cruz Biotechnology, Dallas, TX, USA) overnight at 4 °C. The membranes were treated with antibodies present in the Oxidative Phosphorylation (OXPHOS) rodent WB antibody cocktail (1:1000, #ab110413; Abcam, Cambridge, UK) for samples that underwent mitochondrial isolation under the same conditions. The aforementioned OXPHOS cocktail includes five antibodies, one each against mitochondrial complex I subunit NDUFB8 (ab110242), complex II–30 kDa (ab14714), complex III–Core protein 2 (ab14745), complex IV subunit I (ab14705), and complex V α subunit (ab14748). The membranes were immersed in the TBST buffer containing 5% skim milk and horseradish peroxidase-conjugated goat anti-rabbit or goat anti-mouse immunoglobulin G as the secondary antibody for 1 h. The proteins were detected by ECL Western blotting substrate (Thermo Fisher Scientific, Waltham, MA, USA), and immunoblot bands were checked by an image analyzer system (Syngene, Bethesda, MD, USA). For optic nerve samples, the amount of protein in the optic nerve for each eye was too small, so the total optic nerve tissue for each group was combined into a single sample for each group. Western blotting for anti-Iba-1 in optic nerve samples was performed in the way mentioned above, but statistical analysis could not be completed.

### 4.8. Ultrastructural Examination

The optic nerve tissues were fixed in 2.5% glutaraldehyde at room temperature. After washing them with PBS, samples were placed in 1% fixed osmium tetroxide, dehydrated in alcohol, and embedded in Epon resin. The samples were sectioned and visualized under a transmission electron microscope (JEDM 1010; JEOL, Tokyo, Japan).

### 4.9. Data Analysis

Statistical analyses were carried out using SPSS software (ver. 17.0; IBM Corporation, Armonk, NY, USA). All data are indicated using mean ± standard deviation values. The results with *p* < 0.05 were considered statistically significant. Differences among groups were investigated by one-way analysis of variance. The Turkey’s b post hoc test was performed to determine whether there was a significant difference between groups and to adjust the multiple comparisons after applying one-way analysis of variance. The correlation between variables was established using Pearson’s correlation coefficient.

## 5. Conclusions

This study has demonstrated that IOP fluctuation could be induced by the instillation of IOP-lowering eyedrops in normal rats. Greater IOP fluctuation led to greater oxidative stress and triggered macrogliosis in the retina and the optic nerve head, resulting in the degeneration of RGCs. The maintenance of stable IOP values seems to be critical, and it is advisable to encourage consistent adherence to glaucoma medication, even in patients with NTG.

## Figures and Tables

**Figure 1 ijms-25-03689-f001:**
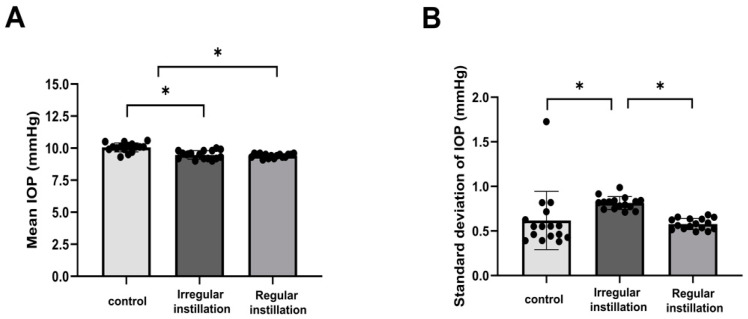
Mean intraocular pressure (IOP) and IOP fluctuation. (**A**). The mean IOP was lower with the irregular instillation of IOP-lowering eyedrops (brinzolamide, latanoprost) than that in the normal control group or with the regular instillation of IOP-lowering eyedrops (* *p* < 0.001). (**B**) The irregular instillation group showed a greater standard deviation of IOP measurements than the normal control group or the regular instillation group (* *p* < 0.001). The mean and error bars indicate the mean and the SD of SD measurements taken from individual eye within each experimental group.

**Figure 2 ijms-25-03689-f002:**
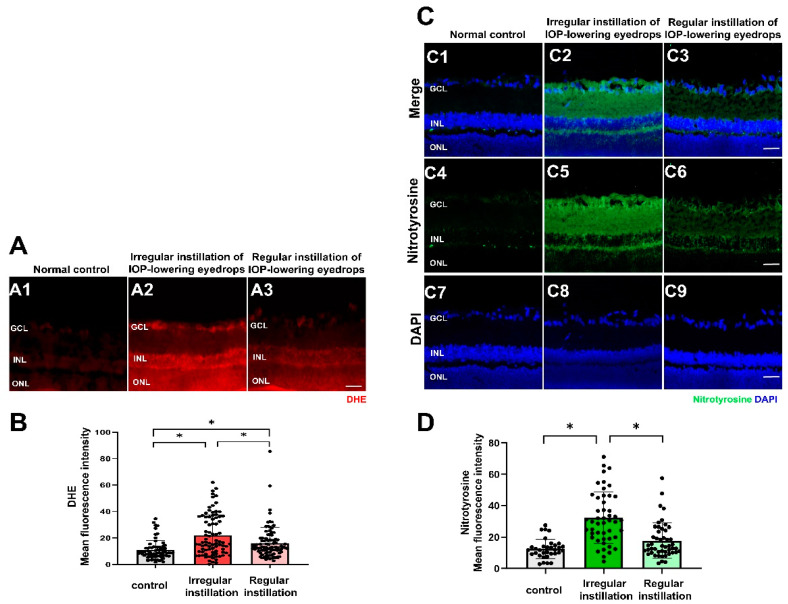
Oxidative stress. (**A**,**B**) Dihydroethidium (DHE) staining, which detects superoxide, revealed increased expression spanning the vertical retinal section, including the ganglion cell layer (GCL) in the group receiving irregular instillation of intraocular pressure-lowering eyedrops (* *p* < 0.001). (**C**,**D**) Immunofluorescence staining for nitrotyrosine, which is a detectable marker of peroxynitrite formation, showed increased expression in the GCL, inner plexiform layer, and outer plexiform layer in the irregular instillation group (* *p* < 0.001). GCL, ganglion cell layer; INL, inner nuclear layer; ONL, outer nuclear layer. Scale bar = 20 μm.

**Figure 3 ijms-25-03689-f003:**
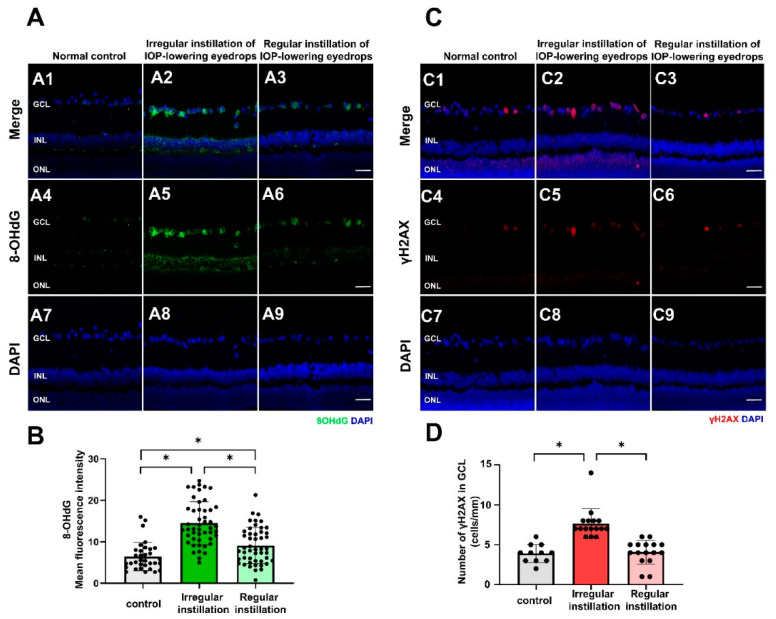
Oxidative DNA damage displayed by 8-hydroxy-2′-deoxyguanosine (8-OHdG) and anti-γH2AX expression. (**A**,**B**) With the irregular instillation of IOP (intraocular pressure)-lowering eyedrops, immunofluorescence staining of 8-OHdG showed increased expression, especially in the ganglion cell layer (GCL), and weakly in the inner nuclear layer (INL) compared to in the normal control group or the regular instillation group (* *p* < 0.001). (**C**,**D**) Anti-γH2AX expression was greater in the GCL in the irregular instillation group than in the normal control group or the regular instillation group (* *p* = 0.014). Scale bar = 20 μm.

**Figure 4 ijms-25-03689-f004:**
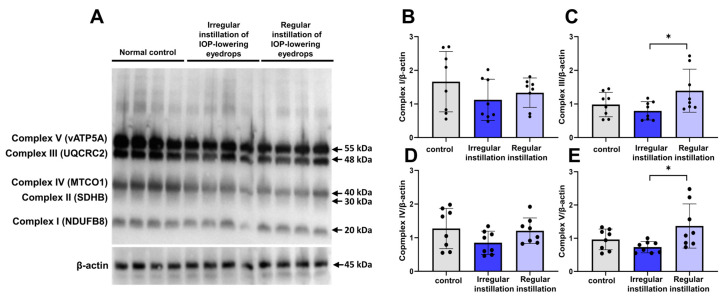
Western blotting for mitochondrial protein for oxidative phosphorylation. (**A**–**E**) The oxidative phosphorylation complexes III and V in mitochondrial samples isolated from the retina displayed a significant difference between eyes with the irregular instillation of intraocular pressure (IOP)-lowering eyedrops and those with the regular instillation of IOP-lowering eyedrops (* *p* = 0.043 and *p* = 0.025, respectively; (**C**,**E**)). However, the irregular instillation group did not show a significant difference compared to the normal control group (according to Turkey’s post hoc tests) in complex III or V expression.

**Figure 5 ijms-25-03689-f005:**
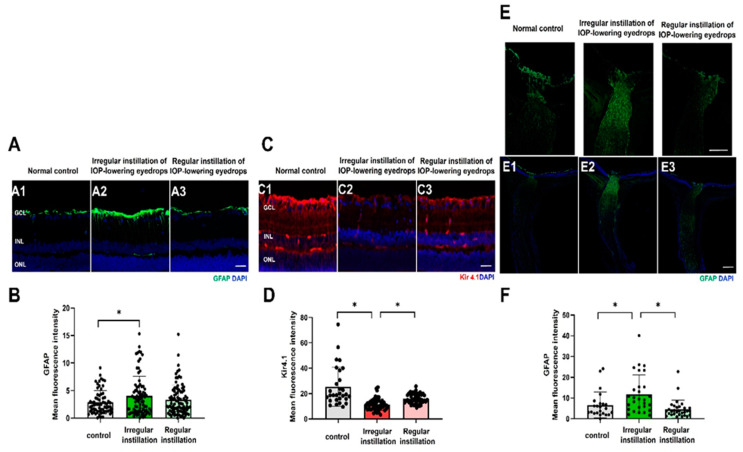
Macroglial activation in the retina and optic nerve head (ONH) and inwardly rectifying Kir channels in the retina. (**A**,**B**). In the normal control group, glial fibrillary acidic protein (GFAP) expression was limited to the astrocytes in the retinal nerve fiber layer and the end-feet of Müller cells at the internal limiting membrane. In the irregular instillation group, GFAP immunostaining was increased to the inner plexiform layer, and the extent of GFAP expression was greater than that in the control group (* *p* = 0.045). (**C**,**D**) Immunofluorescence staining of Kir 4.1 spanned the entire retina representing the Müller cells in the normal control group. Kir 4.1 expression was decreased in the irregular instillation group compared to the normal control group and relatively preserved in the regular instillation group (*p* < 0.001). (**E**,**F**) Regarding the ONH, immunofluorescence staining for GFAP was upregulated in the glial lamina, showing astrogliosis in the irregular instillation group compared to in the normal control group or regular instillation group (* *p* = 0.001). Scale bar = 20 μm (**A**,**C**), 200 μm (**E**).

**Figure 6 ijms-25-03689-f006:**
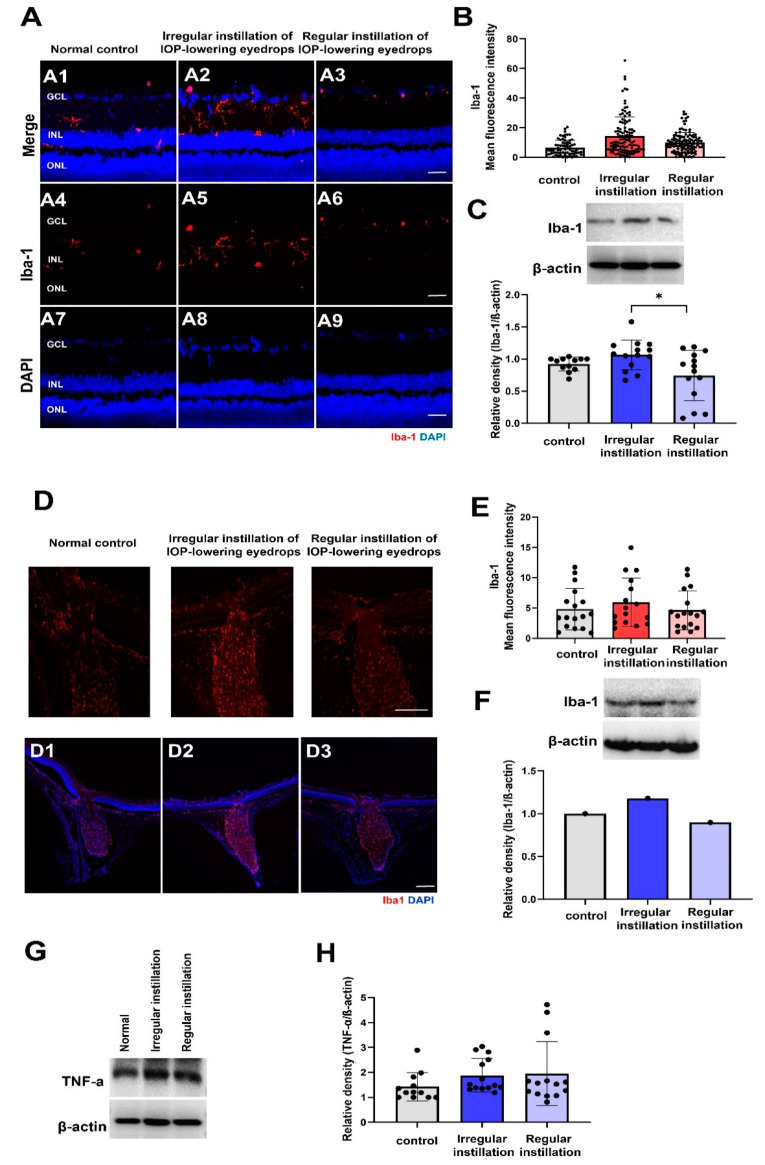
Microgliosis. (**A**,**B**) Immunostaining of Iba-1 showed enlarged soma of microglia, with the irregular instillation of IOP-lowering eyedrops (**A**), but there was no significant difference between groups (**B**). (**C**) The Western blotting results for anti-Iba-1 did not reveal a significant difference between the irregular instillation group and the normal control group (* *p* = 0.05) (**D**,**E**) In the optic nerve head (ONH), immunofluorescence labeling for Iba-1 was increased in the irregular instillation group, but there was no significant difference between the groups (*p* = 0.546). (**F**) Western blotting results of the optic nerve showed a tendency for greater expression of Iba-1 in the irregular instillation group but statistical analysis could not be completed because the optic nerve samples were few in number and were combined for each group. (**G**,**H**) Expression of TNF-α was not different between the groups (*p* = 0.303). Scale bar = 20 μm.

**Figure 7 ijms-25-03689-f007:**
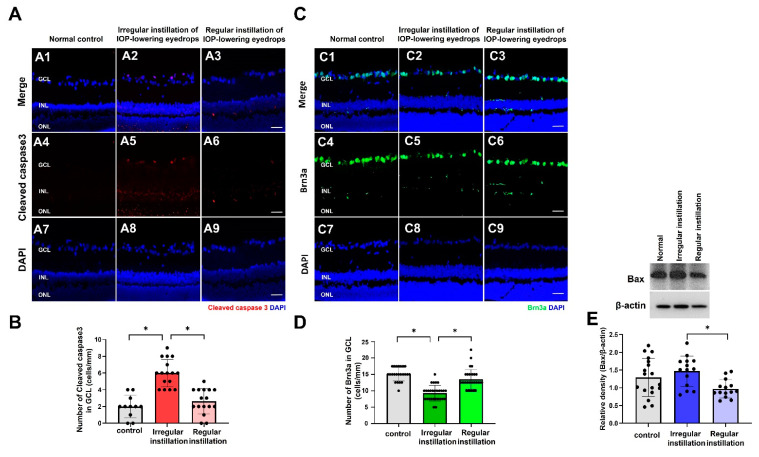
Apoptosis and degeneration of retinal ganglion cells (RGCs). (**A**,**B**) In the irregular instillation group, immunofluorescence staining of cleaved caspase-3 was positive in a few cells in the ganglion cell layer (GCL), and the number of cells stained for cleaved caspase-3 was greater in the irregular instillation group (* *p* = 0.014). (**C**,**D**) The labeling of Brn3a was prominent in the GCL in all groups. The number of Brn3a-positive staining cells in the GCL was lower in the irregular instillation group compared to the normal control group or regular instillation group (* *p* = 0.002). (**E**) Following Western blotting, the irregular instillation group revealed increased expression of Bax compared to the regular instillation group, but there was no significant difference compared to the normal control group (* *p* = 0.014, analysis of variance and Turkey’s post hoc tests). Scale bar = 20 μm.

## Data Availability

Data is contained within the article and Appendix A.

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
