# Peer review of "Retinal Neurodegeneration in an Intraocular Pressure Fluctuation Rat Model"

_ijms, 2024, doi:10.3390/ijms25073689_

Round 1

Reviewer 1 Report

Comments and Suggestions for Authors

Glaucoma, its main causative factor elevated IOP, and its consequences such as retinal degeneration is always in interest of scientific researchers and medical doctors, and the end beneficial that is patients because of the increasing numbers of patients at global level. However, the finished and ongoing researchs provide clues for managing Glucome. In this respect, authors are appreciated to making a study plan, performing study and finaly presenting the results. This paper satisfactorily introduce the objective of the research study, methodology, data generated and results obtained, discussion and conclusions. However, the study is performed with animals, and it is not very clear how this study findings will be applied in clinical practice. Typing errors should be checked such as "instillatioIJMSn". It should be avoid to repeat some words such as "In this study". Refences must be more updated.

Reviewer 2 Report

Comments and Suggestions for Authors

This manuscript describes a new model of IOP fluctuation in male adult Sprague-Dawley rats. The authors provide sufficient data describing important characteristics that are known to happen in glaucoma and multiple glaucoma models. The irregular fluctuation of IOP within normal range, produces RGCs death, oxidative stress, macrogliosis and microgliosis. The manuscript is overall well written with a few redaction and spelling mistakes. Figures content is sufficient, but presentation and quality can improve. 

In Figure 1 the authors present mean and standard deviation of IOP in separate graphs. It is understood that the mean graph presents mean +- standard deviation. But it is not clear what the deviation bar represents in the standard deviation bars, neither it is clear what each observation (point) means in that graph. Are these the deviation of 5 measurements of 16 animals? deviation among the repeated measurements of each animal per day measured? or another parameter? Please include that in the figure legend.  

Authors clarify that for histological analysis, two mid and two peripheral sections of the retina were delimitated. However, authors do not clarify how measurements were made considering these sections. Please specify to what section the representative images correspond to (Fig 2, 3, 5, 6, 7). Please provide more details on how fluorescence quantifications were made and the units (DHE, nitrotyrosine and 8-OHdG, GFAP, Kir 4.1, Iba1). It is not clear if values represent total mean fluorescence intensity, fluorescence per area or another parameter. 

Authors evaluated an irregular instillation group, a regular instillation group and a control group. The irregular instillation group develops detrimental characteristics. However, the regular instillation group develops oxidative stress and shows certain tendencies in the other parameters evaluated. Lowering IOP has shown beneficial effects even in NTG (ex. doi: 10.4103/2008-322X.183914). Yet authors report that lowering IOP to normal animals can cause oxidative stress. Please discuss these results as they are not stated in the discussion. 

Why did the authors choose to make a negative correlation between GFAP/Brn3a? Please provide more information and justify the relation. 

Consider using individual labeling of images of IHC and fluorescence, since it can be difficult to follow the observations stated in the text without having specific panel labeling with each fluorescence, dapi and merge photograph per group. 

Iba1 behavior is contrary to some reports, in which microglial activation causes a shrinking of microglial projections, instead of promoting them (ex. doi: 10.14218/JERP.2020.00015). Please add discussion about that. 

Sex and sex hormones can have a strong effect on inflammation, gliosis, IOP and blood flow (ex. DOI: 10.1097/IJG.0000000000001106). Consider adding the sex of the animals in the title.

A greater magnification of the ONH IHCs of GFAP and Iba1 is suggested since the ones provided are almost the same as the panoramic images. Indicate that the images above are magnifications of the ones below. Add scale bar in the magnifications. And in case of having the images, including GFAP and Iba1 alone without DAPI for a better appreciation. 

It is recommended to re-organize figure 6. To align the IHC in pair with their corresponding WBs in the same line might look better and easier to follow than the current form. Also, why are there IHC quantification of the ONH but not in the retinas? 

As mentioned before, it is recommended to add labeling to individual photographs and panels, so the description of the results is more detailed and easier to locate. Figure 7 is an example of very few labels (just A, B and C) for too many images and information. In the same figure, the graphs show a Y scale that goes from 0.0 to 0.5 to. It is not clear how they reflect number of positive cells to Brn3a or cleaved caspase-3. Is it a proportion of total cells? Is it relative to area? Is it percentage? Please clarify. Also, there are plenty of observations at 0.1 with cleaved caspase 3, what does that mean? If the representative images show no positive sign at all.

The description of Figure S4 states that axoplasm is relatively sparce and is indicated by an asterisk, however no asterisk is found in the figure. 

Regarding the WB. Many antibodies show a lot of positive bands, and the ones chosen are not the ones that correspond to the theorical molecular weight. For instance, Iba1 MW is around 17 kDa, however author’s chosen bands are around 40 kDa in one blot and in the other one, even when lower MW bands are present, authors chose the one around 40 kDa. Please justify this. Same with P2Y12, the predicted MW is 39 kDa, yet you use the 50 kDa band even though there is a ~39 kDa band. TNFa also showed plenty of bands and probably precursor and mature forms, why is the ~25 kDa chosen? Consider adding up all the bands for the analysis if separating the bands is not possible. 

Authors should speculate more in the discussion about the possible mechanisms that lead to all the described results in the irregular instillation group. The discussion around the possible detrimental effects of not following regular schemes of eye drops in glaucoma patients is very important in the clinical practices and making more emphasis in that is recommended (is there any literature that compares strict and non-strict follow ups of treatments in glaucoma or similar?).

Note: the image quality is very low in the PDF provided. Authors should make sure the final version has the best quality of images possible. 
